# Social Connectivity, Sentiment and Participation on Twitter during COVID-19

**DOI:** 10.3390/ijerph18168390

**Published:** 2021-08-08

**Authors:** Andrea Castro-Martinez, Paula Méndez-Domínguez, Aimiris Sosa Valcarcel, Joaquín Castillo de Mesa

**Affiliations:** 1Department of Audiovisual Communication and Advertising, University of Malaga, León Tolstoi Street, s/n, 29010 Malaga, Spain; andreacastro@uma.es; 2Department of Social Psychology, Social Work, Social Anthropology and East Asian Studies, University of Malaga, Francisco Trujillo Villanueva Avn., s/n, 29001 Malaga, Spain; pamendez@uma.es (P.M.-D.); jcastillodemesa@uma.es (J.C.d.M.)

**Keywords:** COVID-19, connectivity, interaction, participation, sentiment, social capital, social media, Twitter

## Abstract

In a transnational context defined by the irruption of COVID-19 and the social isolation it has generated around the world, social networking sites are essential channels for communicating and developing new forms of social coexistence based on connectivity and interaction. This study analyzes the feelings expressed on Twitter through the hashtags #YoMeQuedoEnCasa, #stayhome, #jeresteàlamaison, #restealamaison, #stoacasa, #restaacasa, #ficaemcasa, #euficoemcasa, #ichbleibezuHause and #Bleibzuhause, and the communicative and social processes articulated from network participation, during the lockdown in 2020. Through Gephi software, the aspects underlying the communicative interaction and the distribution of the network at a global level are studied, with the identification of leaderships, communities and connectivity nodes. As a result of this interaction, the emergence of social and organizational links derived from community participation and motivated by the common interest of preserving health and general wellbeing through collective action is detected. The study notes the presence of feelings of solidarity, a sense of community and social support among connected crowds who, despite being in geographically dispersed settings, share similar concerns about the virus effect.

## 1. Introduction

COVID-19 has had a significant global impact not only in health and economic terms but also on the population’s habits and lifestyles, as people have been forced to adapt quickly to an unprecedented global crisis [1]. According to the World Health Organization [2], the first case of the disease was reported on 31 December 2019. By the end of April 2020, more than 3 million cases and 211,000 deaths had been reported [3]. As the virus spread quickly in Spring 2020, which is considered the first wave [4], the WHO advised the population to stay at home, and most countries decreed lockdowns to curb the spread of the disease [5]. Some of the physical and psychological effects caused by this social isolation have included stress, fear, uncertainty, confusion, loss of social relationships, depression, anger, anxiety, and even post-traumatic stress [6,7]. At the beginning of April 2020, over 90 countries or territories comprising half of the world’s population were requested to undergo or forced into quarantine [8].

Any form of socialization was drastically reduced due to the lockdown, which led to increased digital use and consumption to access social relationships, information, and leisure [9,10]. The lack of personal contact gave rise to a trend in digital hyperconnection [11]. Communication plays a vital role in managing any health crisis by helping to control the population’s perception of risk [12].

Social media consumption increased in these circumstances [13] as a form of entertainment and to obtain up-to-date information since, in critical situations, social media is used to obtain information [14]. This phenomenon is also known as an infodemic [15], because false information and fake news about the virus and its effects can spread rapidly and constantly evolve [16]. This situation came about because mobile applications and social networking sites constituted the most significant traffic source to media portals and hosted most of the unreliable news [17].

Social networking sites were essential for disseminating information during the lockdown. The most shared contents were advice and recommendations about the virus, messages of encouragement to healthcare workers and security groups, etc., memes/humor about COVID-19, and to a much lesser extent denunciations of crisis management and of the latest news about the pandemic [18]. Pulido et al. [19] found that although false information is tweeted more than usual, it is retweeted less than scientific or fact-checked publications.

Twitter proved essential for constructing public opinion and disseminating information, as had already been the case with Ebola [20]. The role of Twitter during the pandemic has been the subject of several publications that focus on the topics of conversation [21], feelings generated in specific countries [22,23,24] or institutions’ and organizations’ use [25,26].

### Connectivity, Participation, and Social Capital on Social Networking Sites

A longstanding tradition in Social Science research has shown the implications of interacting for interpersonal networks in creating social ties for citizens to participate in public affairs [27,28,29,30]. This is a subject of study that is of growing interest, especially in recent decades with the development of information and communication technologies. The use of communication technologies via the internet, social networking sites, and digital platforms favor the creating and exchanging of content on a global scale. This consolidates new forms of organization, participation, and political and social mobilization [31].

This paradigm shift towards a model of collective coexistence in the public digital space has created a scenario of social connectivity in which power structures are redefined [32]. This has led to the gestation of collective action through social networking sites, across a range of issues, which amplify the connected multitude’s feelings [33], especially in social emergencies. We agree with Schroeder and Vilo that such actions are “fundamental pillars of citizen participation in the management processes of public space,” as they allow interaction “in an intangible space to form ties that can be transferred to material space” [34] (p. 107).

There is consensus in the scientific community about the role of social networking sites in transitory circumstances caused by adverse events such as natural disasters, crises, or catastrophes that require immediate measures to minimize their consequences [35]. Research that analyzes diverse actors’ uses of these networks provides background information. In emergency scenarios such as those described above, these actors attempt to seek and obtain solidarity and social support through interaction on online social media [36,37,38,39].

Oh et al. [40] identified social support on social networking sites as perception of support and sense of community, which can occur both in everyday situations and crisis moments. As Castillo de Mesa & Gómez-Jacinto [41] highlighted, this fraternal environment not only promotes psychological wellbeing, but also fosters the perception of a sense of community, strengthening participation, communication, and community.

On social networking sites such as Twitter, these feelings emerge from users’ interactions when posting, sharing, or reacting to content [42]. The digital space’s characteristic language is used in that interaction, based on audiovisual and hypertextual resources.

Abilities or behaviors which are used to respond to possible transformations that may arise in an individual’s or collective’s life, associated with social support, solidarity, and a sense of community, are often manifested via social networking sites during traumatic events. Some studies define these behaviors in terms of resilience, which generally expresses how individuals use and adapt their resources to face adversities in hostile environments [43,44].

In adverse circumstances, the capacity to generate social capital on social networking sites is reinforced by the immediacy of exchanging resources in real-time without the physical world’s limitations. Hence, regardless of the use of social networking sites, the benefit of interpersonal contact forged via them, either material, informational, emotional, or in terms of social support, is considered as social capital [45,46].

The degree of overlap between online and offline networks has been verified in several studies. Often, the relationships on offline networks are also found on online platforms [47,48]. Membership of these social networking sites generates a type of social capital that complements the offline network [49], since accessing information increases along with the connections made [50]. Steinfield et al. [51] found a causal relationship between the patterns in the use of social networking sites and the formation of social capital. Their study suggests that the intensity of online relationships can vary depending on their nature, i.e., whether establishing weak or strong ties.

Putnam’s [29] social capital comprises the construction of communities made up of weak or loose ties of disparate users in the online universe. According to Granovetter [52], these ties or bonds promote new connections with people, usually outside local networks. In contrast to loose relationships that serve as sources of information or social connections, we find the phenomenon of social capital binding; this creates strong, reciprocal ties and close connection, which provides emotional support [53].

In the context of social networking sites, connectivity patterns and the nature of communication exchange determine network position and condition the access to information, influence, and leadership in the networks’ social structure [54]. According to Lin [30], this position will depend on the type of online community to which users belong, and whether they are explicit communities that join consciously or implicit ones that join involuntarily due to their social connections.

These are ways of constructing a virtual space for socialization, which hosts specialized communities. The users identify potential contacts of interest and connect to build their networks around common elements shared in offline reality [55,56]. Consequently, there is a type of proximity in which affinity or common interests matter more than geographical proximity.

Some authors also recognize that while social networking sites provide immediate access to information, democratization and the principles of diversity of true quality content are being challenged by the overabundance of information on digital platforms [57]. Partisan news and information and polarized discussion cause users to isolate themselves in echo chambers, arising from their affinities [58].

This phenomenon where users are exposed to content that is consistent with their ideas reinforces existing beliefs, attitudes, and behaviors [59]. Echo chambers form and evolve, constituting a homophilic attitude that leads potentially to the formation of so-called filter bubbles [60]; users are provided with the content they agree with, allowing them to avoid opposing opinions.

This study aims to investigate the aspects underlying Twitter users’ communicative interaction worldwide, concerning the hashtags #YoMeQuedoEnCasa #stayhome, #jeresteàlamaison, #restealamaison, #stoacasa, #restaacasa, #ficaemcasa, #euficoemcasa, #ichbleibezuHause, and #Bleibzuhause, as well as the communicative and social processes based on network participation during the first months of the 2020 Spring lockdown.

For the analysis of these interactions, we start from the following hypotheses:

**H1.** *Users using the analyzed hashtags on Twitter seem to be connected by communities based on different feelings of solidarity and social support*.

**H2.** *The conversations in these communities revolve around social problems or collective actions resulting from the emergence of COVID-19*.

**H3.** *The interaction on Twitter feeds back into resilient behaviors during lockdown*.

Interest in achieving the objective of this work is based on understanding the worldwide use of Twitter as a communication channel in relation to a unique historical moment, such as that of the start of the COVID-19 pandemic. On a practical level, it may be useful to identify behaviours and feelings transmitted through social networks and in the future to determine whether there has been an evolution in their use in the different waves of the COVID-19 crisis. The theoretical implications of this study can be framed as a contribution to the literature on connectivity, social support and the creation of communities in the digital sphere.

## 2. Materials and Methods

### 2.1. Sample

The investigation’s study period took place between 19 March and 19 April 2020, when almost half the world’s population was in lockdown [8]. During this period, five random samples were gathered (Figure 1), one for each week of the period analyzed, from posts on Twitter that used hashtags in different languages. The total sample includes 176,966 publications.

### 2.2. Procedure

Social Big Data makes it possible to analyze millions of data items in real-time but makes it challenging to interpret users’ motivations, behaviors, and attitudes. To balance this deficiency, Wang [61] states that there is a need for Thick Data. This term highlights Geertz’s [62] concept of dense description to analyze phenomena, cultures, and relationships between people. Thick data provides more valuable information, which makes it possible to contextualize and interpret. In the online universe, hashtags make it possible to identify contexts of massive interaction and to catalyze the conversation. Thus, using the Hashtagify tool, the hashtags with the highest level of interaction on a global scale were selected during the dates on which the samples were collected. These were #YoMeQuedoEnCasa, #stayhome, #jeresteàlamaison, #restealamaison, #stoacasa, #restaacasa, #ficaemcasa, #euficoemcasa, #ichbleibezuHause, and #Bleibzuhause.

The following procedures have been followed to verify the hypotheses. First of all, the data was extracted using the Twitter Streaming Importer plugin associated with Twitter’s API. Next, the Force Atlas2 distribution algorithm was applied, which simulates a physical system for distributing the nodes of a network. The profiles are represented as points/nodes and the relationships between the nodes (and the retweets) as connections/lines between the retweeter and the retweeted. The size of the nodes is proportional to the number of retweets received so that the profiles with the most significant impact stand out. The color is assigned according to the community and the artists of the retweeters’ profile, facilitating the interpretation of the data through visual inspection.

On the other hand, different relationship properties were measured, which show the characteristics of the online social structure. On the one hand, degree centrality is the number of actors to which an actor is directly tied [63]. Another analysis was the degree of betweenness of the network nodes, which measures the frequency with which the node appears in the shortest route between the networks’ nodes [63]. In the cohesion analysis, the measure of closeness was considered, which is the shortest route that a node needs to reach other actors in the network [63]. The average distance between the nodes in the five samples has been compared.

In addition, the level of cohesion in the entire network was analyzed using Latapy’s clustering coefficient algorithm [64], whose coefficient applied to a V node indicates the probability that any pair of randomly chosen nodes are neighbors of V and are linked to each other. The algorithm captures how a network has high and low-density zones. It quantifies how intertwined these zones are and individually measures the density of ties between the connected nodes to a specific node on the network [65].

Finally, based on the modularity algorithm, which breaks down the social structure by joining the nodes with more common connections, the communities within the graph are detected and differentiated by color according to their affinities [66]. These methods have been developed in Gephi software [67], version 0.9.2.

Netnography, defined as the set of methods for recording and interpreting digital environments, which attempts to adapt the notions and guidelines of classical ethnography to the new places of technological mediation [68,69], was used to find out on which basis and factors the communities detected on Twitter were formed. This type of ethnography facilitates an approach to the phenomena that derives from the generalization of web 2.0 tools, in which collaborative, cooperative and dialogic dynamics gain a prominent presence and frequency [70].

The analysis of more than 176,000 tweets in the sample delves into aspects underlying the communicative interaction and the distribution of the network at a global level (Table 1).

The netnographic analysis was carried out by three researchers, so in order to unify the criteria to be used it was necessary to prepare a codebook with the aim of maintaining reliability throughout the analysis among the different researchers. A specific design was used to structure this codebook (Table 2). For this purpose, the indicators of social-organizational links have been elaborated, considering conceptualizations of authors such as [71,72,73].

Once the coding system was designed, the same process was followed for each of the five samples: first, the eight communities with the highest presence detected by the software were selected; then the most relevant aspects regarding the organization of each of these communities was checked, based on the reticular properties of their internal network, in order to detect the most relevant nodes and publications; then these highlighted publications were analyzed by applying the codebook to identify the social-organizational links, the topics of conversation and the communicative resources employed; finally, the remainder of the community publications were analyzed and the presence of each of the hashtags checked. This process was followed with each of the 40 communities in the sample. In this way, the ordered classification according to the type of content made it possible to observe and compare the behavioral patterns of the different samples, together with the results related to the analysis of the properties of the network structure.

## 3. Results

### 3.1. Relational Network Properties and Community Detection

The results of the analyzed relational properties defined the morphology of the network linked to the hashtags #YoMeQuedoEnCasa #stayhome, #jeresteàlamaison, #restealamaison, #stoacasa, #restaacasa, #ficaemcasa, #euficoemcasa, #ichbleibezuHause and #Bleibzuhause.

The average degree centrality across all samples (Table 3) evolved from 1.735 to 1.979, indicating in which sample there is a higher or lower level of connectivity. Sample 5 reflects a higher connectivity since the number of actors to which another actor is directly linked is higher than for the other samples. Sample 3 shows the lowest level of connectivity. In this sense, connectivity is important in terms of the scope of social capital, which means that the higher the level of connectivity, the greater the social capital, as well as the greater the popularity and leadership in the network structures.

The analysis of social distances evidences the interval between nodes for the different samples. The average distance in the five samples ranged from 2.04 to 6.37. Unlike samples 4, and 5, the average distance in the remaining samples reached levels that are far from the average distance of 3.43 evidenced in Twitter by [74]. That is, samples 4 and 5 present the shortest paths, not in physical distance, but in the number of jumps required to reach the other actors.

On the other hand, the mean value of the clustering coefficient varied in the five samples collected from 0.169 to 0.223. This value, which can range in a scaled manner between 0 and 1, with the level of network imbrication being higher when it approaches 1, reflected a low level in all five samples. This indicates that the total number of links in the network with respect to the actors is of low density, and therefore the network structure is not highly imbricated at the social and structural level.

Using the statistical technique of modularity, which allows us to identify dense clusters of relationships in broad social networks [66], we obtained a division of the analyzed social structure into communities for each sample (Table 4).

These communities are considered implicit because users are not aware of their membership. The differences between communities are distinguished by colors in the resulting graphs (Figure 2, Figure 3, Figure 4, Figure 5 and Figure 6). The modularity values of the samples analyzed ranged from 0.81 to 0.85, being considered optimal values since the appropriate parameter for this measure should be above 0.3 [66].

### 3.2. Netnographic Analysis of Community Activity

The study of the conversation shows a large subdivision into different sized groups, since 100 communities were found. The analysis of the communities shows variations in the language of the hashtag and reports of the evolution of the disease. However, there were slight differences; it did not advance at the same rate in all countries. At times, these groups use particular axes of conversation in their messages, generally linked to a geographical area or specific interest, sport, humourous content, food, entertainment, etc., which remained framed in the main topics of conversation.

In sample 1 (March 20) the language most used in the eight main communities is Spanish and the hashtag #YoMeQuedoenCasa, as their activity is centered in Spain and Latin America, although there are also publications from Italy, Germany and the USA, among other countries, and texts and hashtags are used in different languages. The social links identified are participation, sense of community, social support and recognition of resilient behaviors, as well as criticism of those who do not adhere to health recommendations. The main topics of conversation focus on solidarity, health, leisure and cultural proposals, and denouncing unsupportive behaviors. In the yellow community (12.62%), messages appealing to encouragement and solidarity prevail, as well as respect for barrier gestures. Of great importance also is the sharing of resources that can be useful, and content related to leisure, such as music, sports (individuals or professionals from different countries), or cooking. Criticism of unsupportive behavior and multiple publications that promote and reward resilience are detected. The green community (4.8%) mostly contains messages referring to solidarity and community support through individual and company actions, such as that of a closed hotel that offers its employees to do shopping for neighbors, and expressions of recognition for health workers. The dissemination of news about COVID-19 and institutional messages from different entities, such as the autonomous government of Madrid, have an important presence. The light blue community (4.78%) is characterized by prevention messages and has a very broad geographic scope, since although it is mainly Spanish-speaking it covers Spain and various American countries. The contents are varied: from humor and culture, such as Argentinian cinema, to videos of Bible verses. Criticism of health restrictions can be detected in different countries. The dark blue community (3.69%) is mainly located in Colombia and includes health recommendations, messages of solidarity and encouragement and playful and informative content. Of particular note are the updates from the mayor of Bogota on staying at home. The orange community (3.3%) features a viral message from Real Madrid encouraging people to stay at home. Recommendations on prevention, many humorous, and data on the pandemic are also disseminated. Publications on leisure, culture and sports are identified, as well as those that promote and recognize resilience. In the fuchsia community (3.13%), support for a request for accommodation from a Spanish nurse who was moving to Madrid to work against the pandemic stands out (Figure 7). Thus, the conversation revolves around solidarity and social support, offering solutions and proposals to network members and fostering resilience. The teal (2.47%) and pink (2.43%) communities show social links related to participation, sense of community, social support and recognition of resilient behaviors. The main topics of conversation focus on the health situation and prevention, leisure and cultural proposals and criticism of individuals who do not respect recommendations.

Sample 2, corresponding to March 27, reflects the predominance of the hashtags #QuedateEnCasa, #YoMeQuedoEnCasa and #ficamecasa, as well as the Spanish and Portuguese languages in the content generated by users. In the eight communities that on this day demonstrated a higher level of connectivity, similar conversation topics were recorded, such as nostalgia for sporting events held in the past (yellow community 9.88%, light blue 5.34% and orange 2.96%), messages of encouragement seeking collective awareness of the importance of implementing health prevention measures and staying at home (yellow community, green 8.85%, dark blue 3.65% and fuchsia 2.91%), topical threads on monitoring the evolution of the pandemic (yellow, green, orange and pink community 2.17%), and online training options in various fields: languages (yellow community), traffic signs (green community), use of digital platforms (orange community), etc.

The contents of these communities show the presence of social links such as solidarity initiatives from the business sector, consisting of the donation of protective equipment for health personnel (green community) (Figure 8), or the financial support of artists such as Antonio Banderas to local institutions in Spain, for the fight against the coronavirus (orange community). There are also examples of social support for infected people, with the collective call to donate blood in Argentina (fuchsia community), or for women victims of gender violence, who see their situation of vulnerability aggravated by the conditions of confinement (yellow community).

However, the presence of echo chambers around political issues is also detected in some of these communities. These function as a kind of filter that concentrates public debate in specific geographic areas and on issues that affect the citizenry. For example, in the light blue community, visibly geolocated in Brazil, the discussion focuses on the Bolsonaro government’s management of the pandemic and its opposition to restrictions and confinements to curb the virus, due to the economic cost of these measures. The same happens in the orange community, located mainly in Spain, whose conversation is limited to a demand to the Spanish executive to implement legislation that allows the suspension during the confinement of the collection of mortgages, electricity, water, etc. (Figure 8). This phenomenon also appears in the dark blue community, located in Latin America, whose axes of conversation focus on the use of the pandemic by the governments of countries such as Cuba, Venezuela and Nicaragua, to restrict freedoms and fundamental rights, in order to exercise and strengthen their de facto power.

In the April 5 sample, out of the 220 communities detected, eight stood out, accounting for 40.55% of the conversation. The yellow community was the largest (8.25%) together with the green community (7.63%). The languages used in these communities are English and Japanese, where the tweets of political figures stand out due to their high impact on the tweeting community (Figure 9). The light blue and dark blue communities accounted for 5.95% and 5.42% of the conversation, respectively; in these communities, Spanish is spoken and the tweets are mostly related to Easter Week in Spain, where images and videos appeared to recreate the tradition. Next come the orange (4.07%), fuchsia (3.44%), turquoise (3.2%) and pink (2.59%) communities where we find entertainment content, movie proposals, music channels, sports challenges, etc., in English and Spanish from the USA and Spain, while in Asia and Latin America the tweets are more informative about the situation and consequences of COVID-19.

In sample 4 (April 7) the communities are more geographically dispersed, as the content comes from Europe, Latin America, Asia and Africa. Several languages were identified, mainly Spanish, English, Portuguese and Japanese, and it was found that the language of a tweet and the hashtags used were often not the same. The most commonly used hashtags are in Spanish and English. The largest community is the yellow community (7.67%), developed in Spain, Central America, South America and the USA. Messages of encouragement and awareness of health measures, health recommendations, leisure and informative updates prevail (Figure 10). Publications unrelated to the health situation that use hashtags to gain notoriety are detected. The green community (7.04%) covers Europe (Spain, France, UK, Ireland and Bulgaria), America (USA, Canada, Argentina), Africa (Ghana, Nigeria) and Asia and the most commonly used languages are English, Spanish, French and Japanese. Their main content are messages of encouragement, solidarity and support, recognition of healthcare workers and awareness of recommendations. The light blue community (6.56%) covers countries as diverse as Malaysia, Indonesia, Japan, Azerbaijan, the USA, Spain, the UK and Ghana, among others, and uses different languages. Contents related to social support and respect for health recommendations stand out, as well as entertainment proposals. Commercial messages unrelated to the pandemic but using hashtags are detected. The dark blue community (5.37%) is mainly located in Latin America and is mostly Spanish-speaking, although there are posts in Italian, English and Portuguese. The contents focus on awareness-raising, expressions of solidarity and social support, health recommendations and leisure and cultural contents. The orange (4.43%), fuchsia (3.57%), lilac (2.98%) and pink (2.49%) communities account for 13.47% of the sample and are mainly spread throughout Europe, America and Africa, with a prevalence of Spanish and English, although other languages such as French, Dutch, Japanese and Italian are also used. Messages of solidarity and social support and recognition of the people who are fighting against the pandemic prevail. Health and awareness-raising recommendations and recreational content are also highlighted.

In sample 5, the yellow (10.43%) and green (10.08%) communities accounted for most of the conversation on April 19. These communities reflect the interaction of Spanish-speaking countries. In Latin America, the measures adopted by governments are shown and non-compliance with them is denounced (Figure 11). In Spain, social support tweets stand out, usually accompanied by images and videos reflecting positive messages such as the recovery of people who have been hospitalized for coronavirus. In the light blue community (6.54%), political positions in Spain in favor of and against government measures and actions stand out. The next four communities, dark blue (3.67%), fuchsia (2.79%), turquoise (2.73%) and pink (2.74%) barely reach 5% of the sample conversation.

Thus, the results show that users of the analyzed hashtags on Twitter were connected during the first phase of the pandemic, forming communities based on different feelings of solidarity and social support. This corroborates hypothesis 1 of this research.

Thus, some messages have a strong presence in specific communities, but their themes and sentiments correspond to other publications in different communities. There are indicators of community participation in the entire sample, such as examples of collaborating in social initiatives and activities and responses to the appeals organized through social networking sites. An illustrative case is the extensive participation in the applause for healthcare workers and the content taken from these social and recreational gatherings organized at windows and balconies. There is much multimedia content about these tributes in different parts of the world and demonstrations of active participation in the community through games, songs, and initiatives to uphold collective morale, which is also linked to resilience. The sense of community is embodied in numerous posts that indicate common emotional ties, reinforcing the identification with and a sense of belonging to a specific territory while giving rise to collective global awareness of the pandemic. Twitter is used as a tool for establishing social contact via interaction concerning any aspect of the situation.

The results demonstrate strong social support in different facets: emotional support is expressed through messages of affection, gratitude, and encouragement, even between users that do not know each other, and mainly to collectives such as those infected and their families, healthcare workers, and essential workers; instrumental support is expressed in numerous ways such as offering help to high-risk groups, mainly elderly or dependent people, both in matters of force majeure and in everyday activities such as shopping; and informational support is manifested through useful-problem solving content. Some users generate and share publications that serve to highlight particular situations, allowing for spontaneous cooperation. Dissemination of media content is also common.

The analysis indicates that health and resources linked to leisure and culture are the main topics. However, there is a prevalence of one of the two in some communities; both directly affect all users since they are all exposed to the virus and lockdown effects. Regardless of the days and communities, the hashtags share hobbies, interests, and lifestyles. Consequently, the topics of conversation detected corroborate hypothesis 2, as the communities focus on social issues and collective actions resulting from the emergence of COVID-19.

There is a feedback process in terms of resilience since users who demonstrate their adaptability are reinforced by positive messages from the community. These messages are disseminated by the majority, encouraging other resilient behaviors that help to manage stressors. Some posts transmit fear, uncertainty, and at times users’ anger, but this occurs to a lesser extent on a global scale. Messages of collective encouragement and funny content are prominent in the sample and generate more interactions than other types of post. Tweeting is a way to express feelings, be recognized, heard, and understood by the community. This shows that hypothesis 3 is fulfilled, because the interaction generated on Twitter feeds back into the resilient behaviours manifested in the situation of confinement.

Individual and collective solidarity is also a recurring theme in the posts, as well as civility and responsibility. Hashtags are used as an example of respecting the rules and also recriminating others’ non-compliance.

Most posts on Twitter use common communicative resources, such as multimedia content, links, emoticons, gifs, and memes. Content production is very high in all cases and is disseminated along with content created by other users. In the case of hashtags, besides those analyzed, there are usually tags linked to the pandemic, geographical zone, or the theme of the shared content (Figure 12).

At the global level, there is no significant dissemination of fake news among the communities, nor the presence of overtly denialist positions, although in some we find the formation of echo chambers around criticism of government measures in various countries and political disputes over the management of the crisis.

In some communities, the tweets with the greatest impact are tied to influential public figures and organizations. Still, it is also usual for anonymous users to produce viral content, so the sender’s identity facilitates the dissemination of messages but is not a defining factor.

## 4. Discussion and Conclusions

The outbreak of COVID-19 and the consequent social isolation that it has caused is reshaping the global population’s behavioral patterns and lifestyles, which is heading towards a scenario of digital connectivity that is not only constant but necessary [1,11]. In these circumstances, social networking sites have emerged as fundamental channels for communicating and developing new ways of coexisting, dialoguing, and participating socially, and accessing leisure and information resources in an adverse context [9]. However, the growing digital hyperconnection caused by the crisis [11] resulted in a situation of infodemia [15] that led to a greater reach of fake news [16] and unreliable information [19].

Twitter, in particular, has played a central role during the entire pandemic as an informative and public-opinion shaping channel [20], which has contributed to managing the population’s perception of risk by managing human interaction in the virtual public space [12]. However, as the crisis has progressed, this platform has also hosted denialist and anti-vaccine discourses [75].

Despite the fact that there is research that points to the expression of negative feelings during the pandemic through Twitter [76,77,78], the results of this study are aligned with other works that impact the possibility of using Twitter constructively during the COVID-19 crisis and report a use of the network incorporating positive and socially supportive values [79,80,81].

The measurements obtained reveal that the network structures of the different samples are not highly intertwined, which could be due to the fact that interactions, being global, appear to be very dispersed. However, it can be observed that as the pandemic progresses and restrictions are applied and the state of alarm spreads throughout the different countries, interaction and connectivity increase in the last two samples and social distances decrease notably. Patterns of connectivity have been reflected in communities with certain similarities such as geographic location and language; this means that global connectivity continues to be dominated by local ties, with certain patterns reflecting affinities and similarities.

As a result of this interaction, specific social and organization links emerged at the beginning of the COVID-19 crisis, motivated by the common interest in conserving health and general wellbeing through collective action. This is how a sense of community, social support, solidarity, and community participation emerges among connected multitudes who share similar concerns regarding the effects of the virus despite being geographically dispersed.

The collective expression of these feelings and collaborative behaviors in an environment of immediacy such as Twitter also led to manifestations of resilience to overcome traumatic circumstances associated with the pandemic and mobilize public and private resources in the shortest possible time.

During the study period, messages from each user community also show the collective desire not to viralize posts that did not contribute anything to the fight against the pandemic and instead to highlight content that gave importance to levels of interdependence in order to help resolve the health crisis. Consequently, a transnational agenda is established, in which topics that contribute to the quality of public deliberation on matters of general interest associated with COVID-19 are found, such as the number of infected or deaths, public health organization recommendations to avoid the propagation of the virus, each countries’ economic measures to face the paralysis of productive activity, or the psychological, cultural and communicative resources that could be used to favor individual and community adaptation to the new living conditions brought about by the pandemic.

This hegemony of collective discourse builds a social capital powerful enough to overshadow, even within a space of social fragmentation such as Twitter, the echo chambers that were formed based on shared beliefs or opinions [82] that polarized the conversation around political issues in regions such as Spain and Latin America. Thus, the hashtags #YoMeQuedoEnCasa, #stayhome, #jeresteàlamaison, #restealamaison, #stoacasa, #restaacasa, #ficaemcasa, #euficoemcasa, #ichbleibezuHause and #Bleibzuhause act as umbrellas for hosting any content during the first phase of the pandemic, since the only real topic of conversation was the common situation that all network users were experiencing. Regardless of geolocalization, these hashtags were used to identify the users that wanted to promote responsible and supportive behaviors, forming global social ties that contributed to the common good.

This study achieves its objectives by exploring aspects underlying Twitter users’ communicative interaction worldwide, concerning the hashtags #YoMeQuedoEnCasa, #stayhome, #jeresteàlamaison, #restealamaison, #stoacasa, #restaacasa, #ficaemcasa, #euficoemcasa, #ichbleibezuHause, and #Bleibzuhause, as well as the communicative and social processes based on network participation during the first months of the 2020 Spring lockdown.

The analysis of connectivity and interactions has corroborated the three hypotheses formulated. The hashtags analysed on Twitter have been used to connect diverse communities of users around different sentiments of solidarity and social support (H1). The social problems and collective actions generated by the emergence of COVID-19 are the main topics of conversation in these communities (H2). Twitter interaction between different users and communities has been fed back and promoted resilient behaviours during the lockdown (H3).

This paper is a theoretical contribution to the literature on the use of digital social networks as tools to build social capital in the digital sphere through the creation of communities based on connectivity, social support and solidarity. In addition, it shows how Twitter can be used in a positive way during crisis scenarios and periods of isolation such as the one produced by COVID-19, by transmitting the conversation and providing solutions to problems that affect the members of the network. On a practical level, this work allows us to delve into the dynamics of community participation that occurred during the first phase of the pandemic, around a sense of collectivity based on respecting prevention measures for the protection of public health at a global level.

The use of digital communication tools during the COVID-19 crisis constitutes a relevant area of study since the pandemic has reshaped the global population’s habits. This work focuses on the lockdown period during the first wave of the disease. Still, future lines of investigation can move in-depth into aspects such as the evolution of digital interaction on social networking sites, the emergence of echo chambers, and the increase in polarization that can help explain phenomena such as the surge in denialism.

## Figures and Tables

**Figure 1 ijerph-18-08390-f001:**
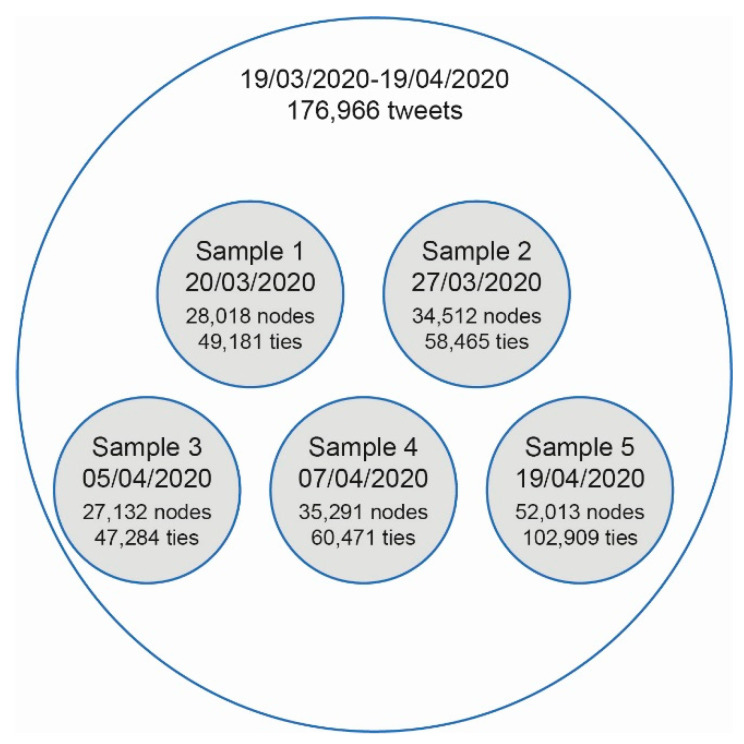
Samples.

**Figure 2 ijerph-18-08390-f002:**
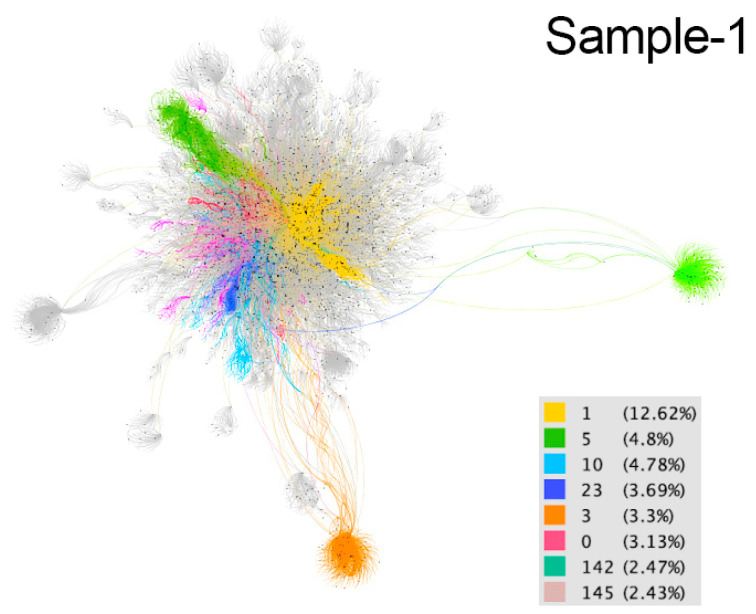
Graph of sample 1. Community detection based on modularity [66] and according to intermediation centrality [63] of sample 1. Source: Gephi [67].

**Figure 3 ijerph-18-08390-f003:**
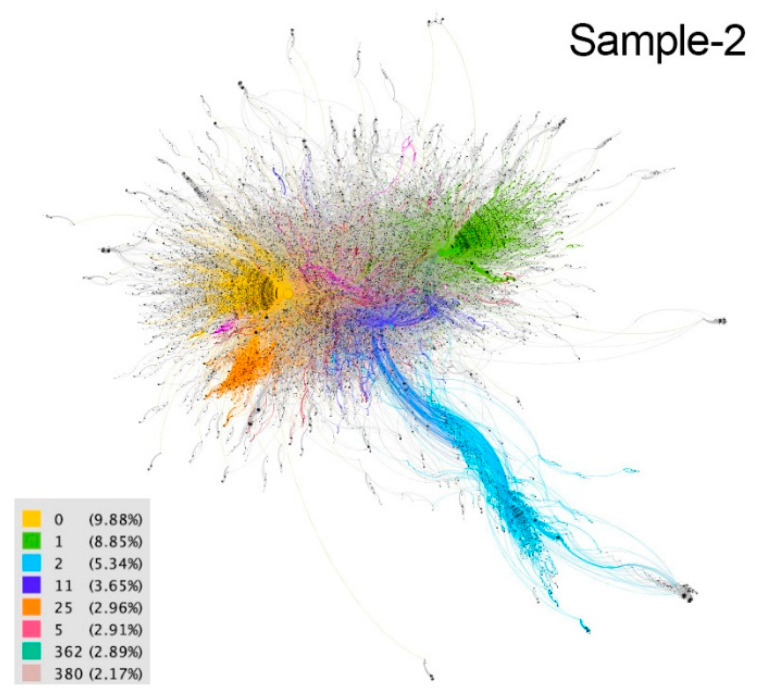
Graph of sample 2. Community detection based on modularity [66] and according to intermediation centrality [63] of sample 1. Source: Gephi [67].

**Figure 4 ijerph-18-08390-f004:**
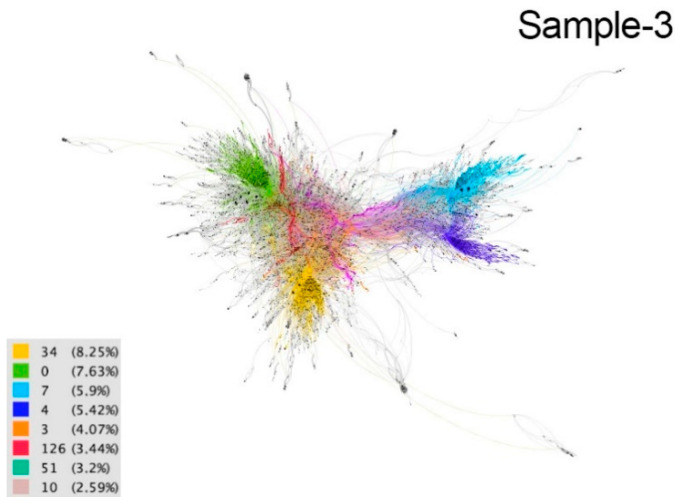
Graph of sample 3. Community detection based on modularity [66] and according to intermediation centrality [63] of sample 1. Source: Gephi [67].

**Figure 5 ijerph-18-08390-f005:**
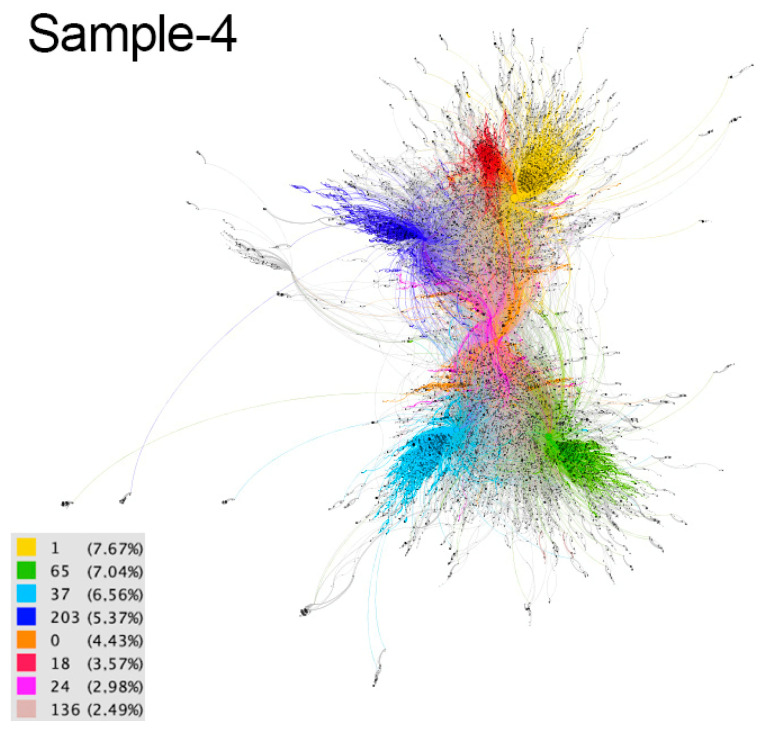
Graph of sample 4. Community detection based on modularity [66] and according to intermediation centrality [63] of sample 1. Source: Gephi [67].

**Figure 6 ijerph-18-08390-f006:**
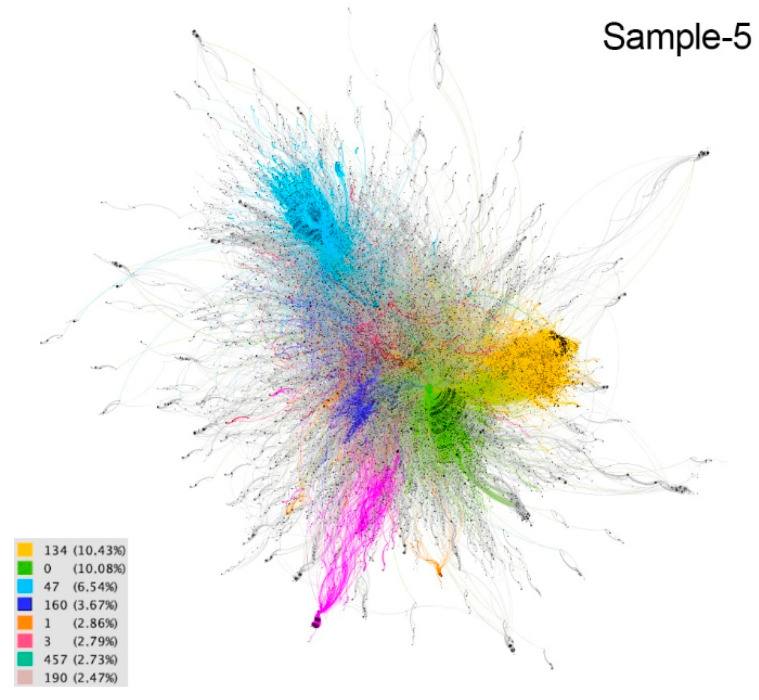
Graph of sample 5. Community detection based on modularity [66] and according to intermediation centrality [63] of sample 1. Source: Gephi [67].

**Figure 7 ijerph-18-08390-f007:**
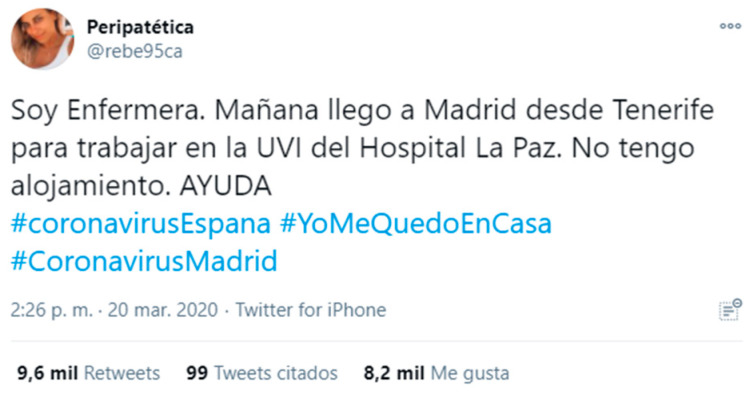
Example of sample content 1.

**Figure 8 ijerph-18-08390-f008:**
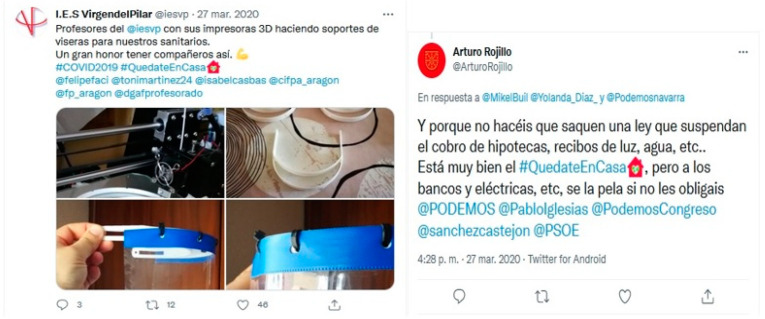
Example of sample content 2.

**Figure 9 ijerph-18-08390-f009:**
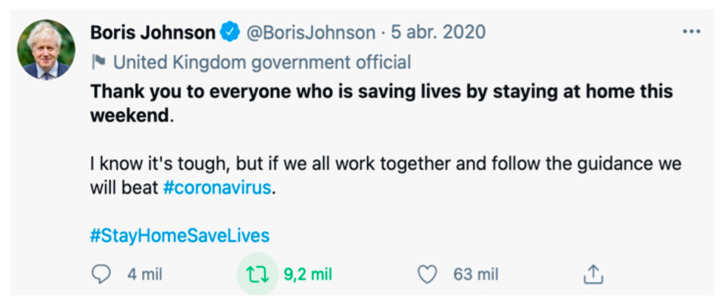
Example of sample content 3.

**Figure 10 ijerph-18-08390-f010:**
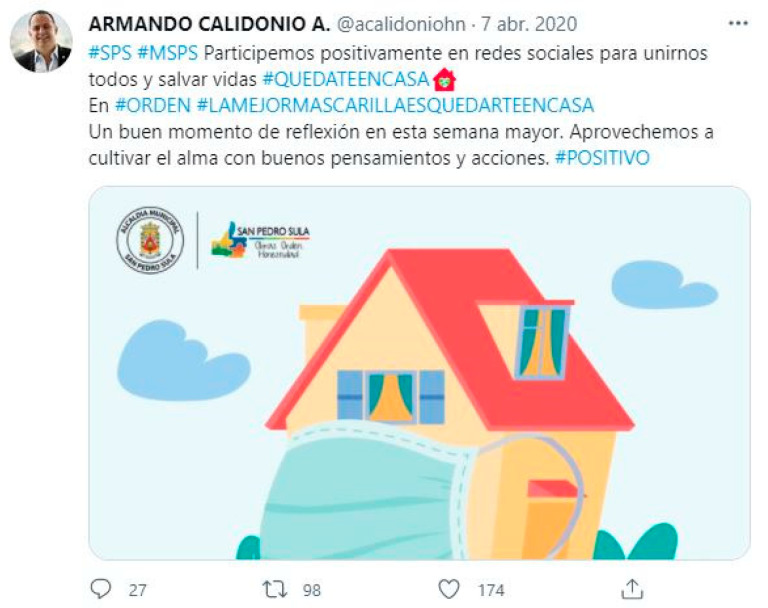
Example of sample content 4.

**Figure 11 ijerph-18-08390-f011:**
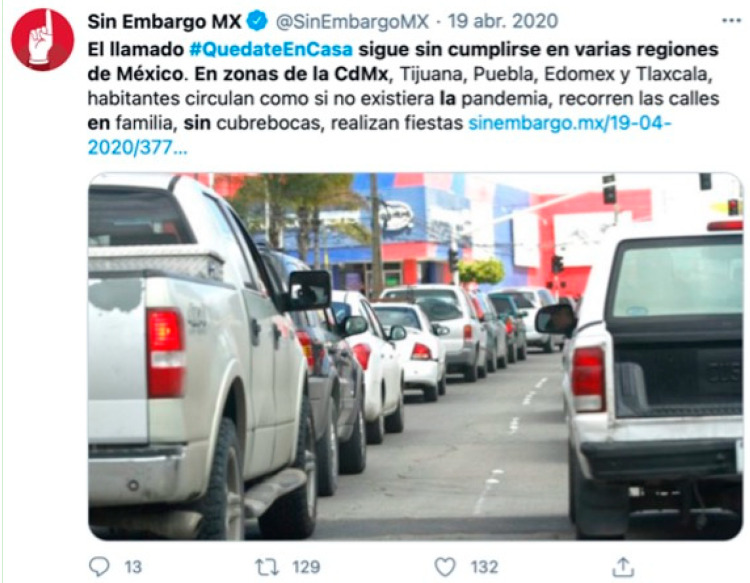
Example of sample content 5.

**Figure 12 ijerph-18-08390-f012:**
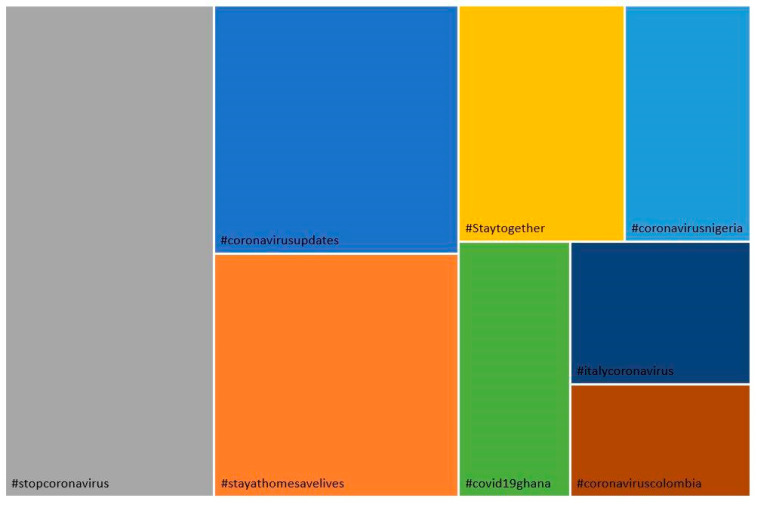
Other hashtags used in tweets.

**Table 1 ijerph-18-08390-t001:** Research design.

Analysis of connectivity and interaction on Twitter around COVID-19	Online social network analysis	Network lattice properties	Degree centrality
			Closeness centrality
			Communities
			Clustering
			Average path lenght
		Community detection	Modularity
	Netnographic analysis of community activity	Axes of conversation	Hashtags
			Topics
		Links	Social
			Organisational

**Table 2 ijerph-18-08390-t002:** Coding book used for netnographic analysis.

Sample	Community	- Identification of leadership - Use of hashtags according to language
Social organizational links	Community participation	- Collaboration with associations-organizations - Participation on social networking sites- activities- initiatives- social demands recreational activities - Responding to the appeals on the networks
	Sense of community	- Satisfaction of needs - Personal link- belonging - Assessment of personal opinion - Help and collaboration - Contact via social networking sites
	Social Support	-Emotional- instrumental- informational
	Resilience	- Adaptability, recovery, and mood - Acceptance of new normal - Collective and communicative learning - Belief that difficulties can make you stronger - Ability to cope with negative feelings - Humor when faced with problems
Topics of conversation	Health	- Advice about COVID - Proposals for physical and mental health
	Entertainment and culture	- Proposal and leisure- entertainment-culture
	Political proposals	- Assessments and policy positions - Political proposals
	Solidarity	- Organized solidarity initiatives - Examples of individual solidarity
	Political criticism	- Positions against the measures adopted - Criticism of the parties and their leaders
	Community criticism	- Denouncement of unsupportive behavior - Criticism of specific communities’ behavior
	Other topics	
Communicative resources	Hashtags	- Presence (high-medium-low)- Most used
	Photos Videos Links	- Presence (high-medium-low) - Source: media/social media/others
Fake news	Fake news	
Leaders-nodes	Popularity- content creation- intermediation	
Echo chambers	Filter bubbles	

Resource: Own elaboration based on the conceptualisations of other authors [71,72,73].

**Table 3 ijerph-18-08390-t003:** Social network analysis.

	Sample-1	Sample-2	Sample-3	Sample-4	Sample-5
Degree centrality	1.735	1.694	1.643	1.743	1.979
Closeness centrality	4.02	2.044	5.181	5.249	5.77
Clustering	0.223	0.214	0.211	0.215	0.169
Average path lenght	5.18	5.24	6.37	2.19	2.04

**Table 4 ijerph-18-08390-t004:** Community analysis.

	Sample-1	Sample-2	Sample-3	Sample-4	Sample-5
Communities	8/521	8/391	8/241	8/265	8/462
Modularity	0.83	0.84	0.85	0.81	0.82

## Data Availability

The data presented in this study were extracted using the Twitter Streaming Importer plugin and they are available through the Twitter API.

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
