# Peer review of "Social Connectivity, Sentiment and Participation on Twitter during COVID-19"

_ijerph, 2021, doi:10.3390/ijerph18168390_

Round 1
Reviewer 1 Report
The article analyzes a topic of great interest. I congratulate the authors for their good job.
However, I have some recommendations about some issues that I think the authors should consider.
The one that I think is the most important is that the article presents a very high volume of information that is not analyzed in depth later.
On the one hand, in table 1 we have a content analysis proposal that is not conducted.
On the other hand, in Table 2 we have some results that may be interesting, but their implications are not analyzed in detail.
I think that the authors have two options: one option would be to expand the length of the article and present all the results together in an expanded way; the other option would be to focus only on one of the two analyzes (either the content analysis from Table 1; or an interpretation of the metrics from Table 2).
Here are these expanded ideas and some other observations:
1. Structure of the article.
1.1. The objective of the article is set out in lines 67-70.
It is recommended to place it at the end of the introductory section, just before the materials and methods section.
Likewise, it is recommended to write a paragraph explaining the interest and need to achieve this objective, as well as the practical and / or theoretical repercussions that may arise from the results.
1.2. The hypotheses are located in the Procedure subsection.
It is recommended to relocate this content at the end of the introduction, after the aim of the paper is exposed.
2. Interpretation of the results.
2.1. Lines 227-229 state "which indicates on the one hand in which sample there is a higher level of connectivity and on the other hand that the level of structure cohesion, even for global networks, was optimal in evaluative terms of the samples".
It must be explained based on what scales or parameters it is argued that these values are "high" in the connectivity variable and are "optimal" in structure cohesion.
2.2. On line 314 a "conversation analysis" is mentioned.
It must be specified by which instruments and criteria this analysis was carried out.
Is it an analysis made from the categories included in table 1?
My recommendation is that either this mention of the "conversation analysis" be removed, or the results of this content analysis be explained in detail.
2.3. Line 332 indicates that "Fake news is disseminated little among the communities worldwide".
What is this claim based on? This is a statement contrary to the evidence from many other researches.
Fake news exist and are being a problem during the pandemic. They probably circulate more through other hashtags not discussed in this article, but they certainly do exist.
2.4. Content analysis.
In lines 254-289 numerous statements are made about the content of the tweets and some illustrative cases are exposed.
It seems that these statements are derived from a content analysis, but there is no sign of systematicity or evidence of validity about this analysis.
Who did it ?; With what criteria ?; Those in table 1?, (if so, it must be specified); Was there an inter-judge reliability procedure? (if so, it should be indicated).
3. Analysis.
3.1. Table 1.
Table 1 presents different categories and indicators to analyze the tweets. It looks like a well-crafted and interesting table.
However, these indicators are subsequently not applied to the results, or they are not explicitly applied.
My recommendation is that either this analysis be done with at least some of the tweet samples, or that the table be eliminated.
3.2. Figure 2.
One of my main concerns is that hashtags in different languages are analyzed together.
It seems logical to think that Twitter users will routinely post their Tweets only in their native language, or in some cases in their native language and a second language like English.
This can clearly influence many of the values included in table 2 and especially in figure 2.
That is, it seems logical to think that the different colors in figure 2 correspond to the different languages of the different hashtags. Isn't it?
If this is so, I think this detracts from the results in Figure 2, as it would be an obvious result.
My recommendation is that you try to redo this analysis, at least for one of the tweet samples, in each of the different languages. This would allow conclusions to be drawn about the degree of cohesion of the communities of speakers of the same language.
4. The discussion must clearly be expanded.
4.1. The results in Table 2 are not actually discussed: their importance, their significance, or their implications are not explained, nor is it compared with the results of other previous studies.
In the discussion, and actually throughout the article, a "friendly" version of Twitter is drawn.
The hashtags that have been selected seem, in fact, that they contain mostly constructive messages that incite the population to join individual efforts for the collective good.
However, there is another, much less constructive side to Twitter. The fake news exist, the echo cambers exist and the messages interested in the individual interest to the detriment of the collective interest exist. This negative version of Twitter cannot be ignored or undervalued.
4.2. I think the discussion should end with a summary of the main theoretical and / or practical conclusions of the results.
Mainly, it should answer the question of, what now? What use can we make of these results? Or, alternatively, what can we learn from these results?
5. Formal issues.
5.1. Keywords must be in alphabetic order.
5.2. There are difficulties in reading the content of the tweets included in the different figures. It is recommended to review it.
Author Response
Dear Reviewer.
In the attached document you will find the responses to the proposed changes. In the new version of the manuscript we have underlined in yellow color all the added information so that you can easily find it.
Please see the attachment.
Kind regards.

Reviewer 2 Report
The manuscript addresses an interesting topic on social connectivity and participation on Twitter during the pandemics of COVID-19. Detailed comments follow:
- The connection between the theoretical part of the manuscript and the empirical part of the manuscript is not sufficient. The lever for the empirical part is not well justified.
- It is not well justified how the hashtags were selected. I recommend adding an appropriate reference to literature. Please, add an explanation why these languages were selected and another languages were dismissed.
- It is not clear how community analysis template is created. I would recommend elaboration of the table 1 along with references from the literature. How indicators were developed?
- I recommend adding a content on related works.
- Figure 3, 4 and 5 are difficult to read. I recommend changing the figures.
- I recommend dropping out the figure 6.
- In the procedure section, we can read about hypotheses, but later on there is a lack of discussion in the results section.
- Discussion is quite poor. I recommend elaborating the discussion section with referring to related works in a greater extent.
- Unfortunately, there are not evident original scientific contributions to a sufficient extent. Practical and theoretical implications of the study are not justified.
Author Response

(The authors gave the same response as above.)

Round 2
Reviewer 1 Report
I think the authors have done a good job. The modifications correctly satisfy the suggestions I made in the first report.
Author Response
Dear reviewer:
We appreciate your kind reply.
Best regards.
Reviewer 2 Report
I commend the authors for significant improvements. I would only suggest the auhors in the final check to reconsider whether commas or dots are appropriately and consistently used for decimal numbers.
Author Response
Dear reviewer:
We appreciate your kind reply.
According to international standards, it is correct to present decimal numbers with both commas and dots. We have considered using commas, because this is the most common standard in Europe.
In the latest version of the manuscript you will find that all decimal numbers are expressed with commas.
Best regards.